# SKIP RNN: LEARNING TO SKIP STATE UPDATES IN RECURRENT NEURAL NETWORKS

**Víctor Campos**[*][†]**, Brendan Jou**[‡]**, Xavier Giró-i-Nieto**[§]**, Jordi Torres**[†]**, Shih-Fu Chang**[Γ]

[†]Barcelona Supercomputing Center, [‡]Google Inc,

[§]Universitat Politècnica de Catalunya, [Γ]Columbia University

```
{victor.campos, jordi.torres}@bsc.es,bjou@google.com,
xavier.giro@upc.edu,shih.fu.chang@columbia.edu
```

## ABSTRACT

Recurrent Neural Networks (RNNs) continue to show outstanding performance in sequence modeling tasks. However, training RNNs on long sequences often face challenges like slow inference, vanishing gradients and difficulty in capturing long term dependencies. In backpropagation through time settings, these issues are tightly coupled with the large, sequential computational graph resulting from unfolding the RNN in time. We introduce the Skip RNN model which extends existing RNN models by learning to skip state updates and shortens the effective size of the computational graph. This model can also be encouraged to perform fewer state updates through a budget constraint. We evaluate the proposed model on various tasks and show how it can reduce the number of required RNN updates while preserving, and sometimes even improving, the performance of the baseline RNN models. Source code is publicly available at https://imatge-upc.github.io/skiprnn-2017-telecombcn/.

## 1 INTRODUCTION

Recurrent Neural Networks (RNNs) have become the standard approach for practitioners when addressing machine learning tasks involving sequential data. Such success has been enabled by the appearance of larger datasets, more powerful computing resources and improved architectures and training algorithms. Gated units, such as the Long Short-Term Memory (Hochreiter & Schmidhuber, 1997) (LSTM) and the Gated Recurrent Unit (Cho et al., 2014) (GRU), were designed to deal with the vanishing gradients problem commonly found in RNNs (Bengio et al., 1994). These architectures have been popularized, in part, due to their impressive results on a variety of tasks in machine translation (Bahdanau et al., 2015), language modeling (Zaremba et al., 2015) and speech recognition (Graves et al., 2013).

Some of the main challenges of RNNs are in their training and deployment when dealing with long sequences, due to their inherently sequential behaviour. These challenges include throughput degradation, slower convergence during training and memory leakage, even for gated architectures (Neil et al., 2016). Sequence shortening techniques, which can be seen as a sort of conditional computation (Bengio et al., 2013; Bengio, 2013; Davis & Arel, 2013) in time, can alleviate these issues. The most common approaches, such as cropping discrete signals or reducing the sampling rate in continuous signals, are based on heuristics and can be suboptimal. In contrast, we propose a model that is able to learn which samples (i.e., elements in the input sequence) need to be used in order to solve the target task. Consider a video understanding task as an example: scenes with large motion may benefit from high frame rates, whereas only a few frames are needed to capture the semantics of a mostly static scene.

The main contribution of this work is a novel modification for existing RNN architectures that allows them to skip state updates, decreasing the number of sequential operations performed, without requiring any additional supervision signal. This model, called Skip RNN, adaptively determines whether the state needs to be updated or copied to the next time step. We show how the network can

---

[*]Work done while Víctor Campos was a visiting scholar at Columbia University.

be encouraged to perform fewer state updates by adding a penalization term during training, allowing us to train models under different computation budgets. The proposed modification can generally be integrated with any RNN and we show, in this paper, implementations with well-known RNNs, namely LSTM and GRU. The resulting models show promising results on a series of sequence modeling tasks. In particular, we evaluate the proposed Skip RNN architecture on six sequence learning problems: an adding task, sine wave frequency discrimination, digit classification, sentiment analysis in movie reviews, action classification in video, and temporal action localization in video[1].

## 2 RELATED WORK

Conditional computation has been shown to gradually increase model capacity without proportional increases in computational cost by exploiting certain computation paths for each input (Bengio et al., 2013; Liu & Deng, 2017; Almahairi et al., 2016; McGill & Perona, 2017; Shazeer et al., 2017). This idea has been extended in the temporal domain, such as in learning how many times an input needs to be "pondered" before moving to the next one (Graves, 2016) or designing RNN architectures whose number of layers depend on the input data (Chung et al., 2017). Other works have addressed time-dependent computation in RNNs by updating only a fraction of the hidden states based on the current hidden state and input (Jernite et al., 2017), or following periodic patterns (Koutnik et al., 2014; Neil et al., 2016). However, due to the inherently sequential nature of RNNs and the parallel computation capabilities of modern hardware, reducing the size of the matrices involved in the computations performed at each time step generally has not accelerated inference as dramatically as hoped. The proposed Skip RNN model can be seen as form of conditional computation in time, where the computation associated to the RNN updates may or may not be executed at every time step. This idea is related to the UPDATE and COPY operations in hierarchical multiscale RNNs (Chung et al., 2017), but applied to the whole stack of RNN layers at the same time. This difference is key to allowing our approach to skip input samples, effectively reducing sequential computation and shielding the hidden state over longer time lags. Learning whether to update or copy the hidden state through time steps can be seen as a learnable Zoneout mask (Krueger et al., 2017) which is shared between all the units in the hidden state. Similarly, it can be interpreted as an input-dependent recurrent version of stochastic depth (Huang et al., 2016).

Selecting parts of the input signal is similar in spirit to the hard attention mechanisms that have been applied to image regions (Mnih et al., 2014), where only some patches of the input image are attended in order to generate captions (Xu et al., 2015) or detect objects (Ba et al., 2014). Our model can be understood as generating a hard temporal attention mask on-the-fly given previously seen samples, deciding which time steps should be attended and operating on a subset of input samples. Subsampling input sequences has been explored for visual storylines generation (Sigurdsson et al., 2016b), although jointly optimizing the RNN weights and the subsampling mechanism is often computationally infeasible and they use an Expectation-Maximization algorithm instead. Similar research has been conducted for video analysis tasks, discovering minimally needed evidence for event recognition (Bhattacharya et al., 2014) and training agents that decide which frames need to be observed in order to localize actions in time (Yeung et al., 2016; Su & Grauman, 2016). Motivated by the advantages of training recurrent models on shorter subsequences, efforts have been conducted on learning differentiable subsampling mechanisms (Raffel & Lawson, 2017), although the computational complexity of the proposed method precludes its application to long input sequences. In contrast, our proposed method can be trained with backpropagation and does not degrade the complexity of the baseline RNNs.

Accelerating inference in RNNs is difficult due to their inherently sequential nature, leading to the design of Quasi-Recurrent Neural Networks (Bradbury et al., 2017) and Simple Recurrent Units (Lei & Zhang, 2017), which relax the temporal dependency between consecutive steps. With the goal of speeding up RNN inference, LSTM-Jump (Yu et al., 2017) augments an LSTM cell with a classification layer that will decide how many steps to jump between RNN updates. Despite its promising results on text tasks, the model needs to be trained with REINFORCE (Williams, 1992), which requires defining a reasonable reward signal. Determining these rewards are non-trivial and may not necessarily generalize across tasks. Moreover, the number of tokens read between jumps,

---

[1]Experiments on sine wave frequency discrimination, sentiment analysis in movie reviews and action classification in video are reported in the appendix due to space limitations.

the maximum jump distance and the number of jumps allowed all need to be chosen in advance. These hyperparameters define a reduced set of subsequences that the model can sample, instead of allowing the network to learn any arbitrary sampling scheme. Unlike LSTM-Jump, our proposed approach is differentiable, thus not requiring any modifications to the loss function and simplifying the optimization process, and is not limited to a predefined set of sample selection patterns.

## 3 MODEL DESCRIPTION

An RNN takes an input sequence $\mathbf{x} = (x_1, \ldots, x_T)$ and generates a state sequence $\mathbf{s} = (s_1, \ldots, s_T)$ by iteratively applying a parametric state transition model $S$ from $t = 1$ to $T$:

$$s_t = S(s_{t-1}, x_t) \tag{1}$$

We augment the network with a binary *state update gate*, $u_t \in \{0, 1\}$, selecting whether the state of the RNN will be updated ($u_t = 1$) or copied from the previous time step ($u_t = 0$). At every time step $t$, the probability $\tilde{u}_{t+1} \in [0, 1]$ of performing a state update at $t + 1$ is emitted. The resulting architecture is depicted in Figure 1 and can be characterized as follows:

$$u_t = f_{binarize}(\tilde{u}_t) \tag{2}$$
$$s_t = u_t \cdot S(s_{t-1}, x_t) + (1 - u_t) \cdot s_{t-1} \tag{3}$$
$$\Delta\tilde{u}_t = \sigma(W_p s_t + b_p) \tag{4}$$
$$\tilde{u}_{t+1} = u_t \cdot \Delta\tilde{u}_t + (1 - u_t) \cdot (\tilde{u}_t + \min(\Delta\tilde{u}_t, 1 - \tilde{u}_t)) \tag{5}$$

where $W_p$ is a weights vector, $b_p$ is a scalar bias, $\sigma$ is the sigmoid function and $f_{binarize} : [0, 1] \to \{0, 1\}$ binarizes the input value. Should the network be composed of several layers, some columns of $W_p$ can be fixed to 0 so that $\Delta\tilde{u}_t$ depends only on the states of a subset of layers (see Section 4.3 for an example with two layers). We implement $f_{binarize}$ as a deterministic step function $u_t = $ round($\tilde{u}_t$), although a stochastic sampling from a Bernoulli distribution $u_t \sim $ Bernoulli($\tilde{u}_t$) would be possible as well.

The model formulation encodes the observation that the likelihood of requesting a new input to update the state increases with the number of consecutively skipped samples. Whenever a state update is omitted, the pre-activation of the state update gate for the following time step, $\tilde{u}_{t+1}$, is incremented by $\Delta\tilde{u}_t$. On the other hand, if a state update is performed, the accumulated value is flushed and $\tilde{u}_{t+1} = \Delta\tilde{u}_t$.

The number of skipped time steps can be computed ahead of time. For the particular formulation used in this work, where $f_{binarize}$ is implemented by means of a rounding function, the number of skipped samples after performing a state update at time step $t$ is given by:

$$N_{skip}(t) = \min\{n : n \cdot \Delta\tilde{u}_t \geq 0.5\} - 1 \tag{6}$$

where $n \in \mathbb{Z}^+$. This enables more efficient implementations where no computation at all is performed whenever $u_t = 0$. These computational savings are possible because $\Delta\tilde{u}_t = \sigma(W_p s_t + b_p) = \sigma(W_p s_{t-1} + b_p) = \Delta\tilde{u}_{t-1}$ when $u_t = 0$ and there is no need to evaluate it again, as depicted in Figure 1d.

There are several advantages in reducing the number of RNN updates. From the computational standpoint, fewer updates translates into fewer required sequential operations to process an input signal, leading to faster inference and reduced energy consumption. Unlike some other models that aim to reduce the average number of operations per step (Neil et al., 2016; Jernite et al., 2017), ours enables skipping steps completely. Replacing RNN updates with copy operations increases the memory of the network and its ability to model long term dependencies even for gated units, since the exponential memory decay observed in LSTM and GRU (Neil et al., 2016) is alleviated. During training, gradients are propagated through fewer updating time steps, providing faster convergence in some tasks involving long sequences. Moreover, the proposed model is orthogonal to recent advances in RNNs and could be used in conjunction with such techniques, e.g. normalization (Cooijmans et al., 2017; Ba et al., 2016), regularization (Zaremba et al., 2015; Krueger et al., 2017), variable computation (Jernite et al., 2017; Neil et al., 2016) or even external memory (Graves et al., 2014; Weston et al., 2014).

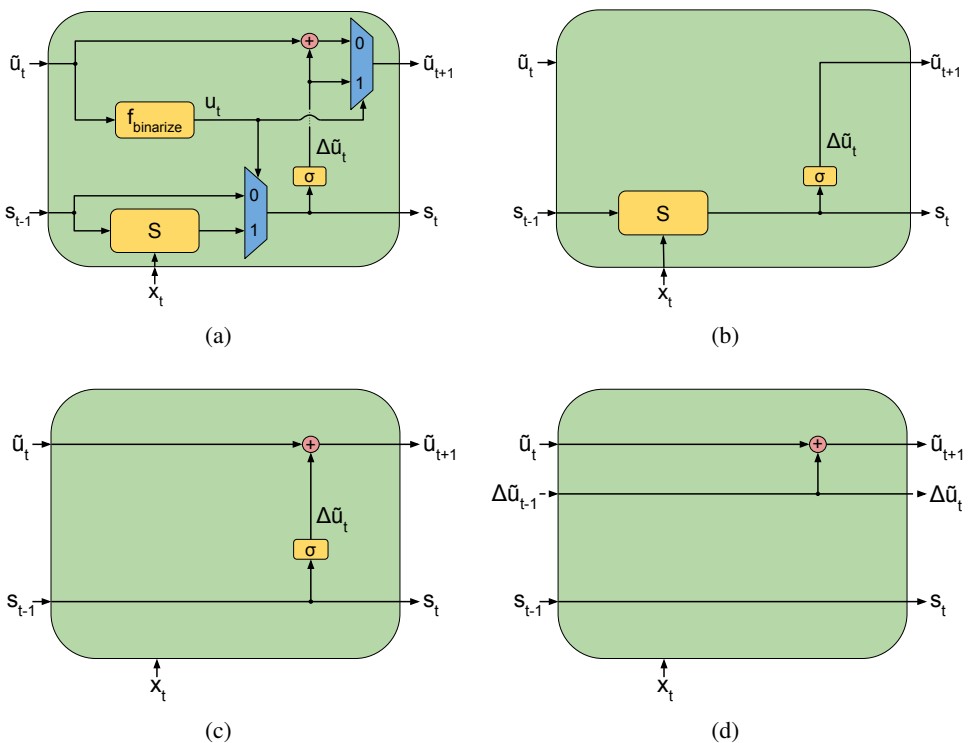

Figure 1: Model architecture of the proposed Skip RNN. **(a)** Complete Skip RNN architecture, where the computation graph at time step $t$ is conditioned on $u_t$. **(b)** Architecture when the state is updated, i.e. $u_t = 1$. **(c)** Architecture when the update step is skipped and the previous state is copied, i.e. $u_t = 0$. **(d)** In practice, redundant computation is avoided by propagating $\Delta\tilde{u}_t$ between time steps when $u_t = 0$.

### 3.1 ERROR GRADIENTS

The whole model is differentiable except for $f_{binarize}$, which outputs binary values. A common method for optimizing functions involving discrete variables is REINFORCE (Williams, 1992), although several estimators have been proposed for the particular case of neurons with binary outputs (Bengio et al., 2013). We select the straight-through estimator (Hinton, 2012; Bengio et al., 2013), which consists of approximating the step function by the identity when computing gradients during the backward pass:

$$\frac{\partial f_{binarize}(x)}{\partial x} = 1 \tag{7}$$

This yields a biased estimator that has proven more efficient than other unbiased but high-variance estimators such as REINFORCE (Bengio et al., 2013) and has been successfully applied in different works (Courbariaux et al., 2016; Chung et al., 2017). By using the straight-through estimator as the backward pass for $f_{binarize}$, all the model parameters can be trained to minimize the target loss function with standard backpropagation and without defining any additional supervision or reward signal.

### 3.2 LIMITING COMPUTATION

The Skip RNN is able to learn when to update or copy the state without explicit information about which samples are useful to solve the task at hand. However, a different operating point on the trade-off between performance and number of processed samples may be required depending on the application, e.g. one may be willing to sacrifice a few accuracy points in order to run faster on machines with a low computational power, or to reduce energy impact on portable devices. The

proposed model can be encouraged to perform fewer state updates through additional loss terms, a common practice in neural networks with dynamically allocated computation (Liu & Deng, 2017; McGill & Perona, 2017; Graves, 2016; Jernite et al., 2017). In particular, we consider a *cost per sample* condition

$$L_{budget} = \lambda \cdot \sum_{t=1}^{T} u_t,$$

(8)

where $L_{budget}$ is the cost associated to a single sequence, $\lambda$ is the cost per sample and $T$ is the sequence length. This formulation bears a similarity to weight decay regularization, where the network is encouraged to slowly converge toward a solution where the norm of the weights is small. Similarly, in this case the network is encouraged to converge toward a solution where fewer state updates are required.

Although the above budget formulation is extensively studied in our experiments, other budget loss terms can be used depending on the application. For instance, a specific number of samples may be encouraged by applying a $L_1$ or $L_2$ loss between the target value and the number of updates per sequence, $\sum_{t=1}^{T} u_t$.

## 4 EXPERIMENTS

In the following section, we investigate the advantages of adding this state skipping to two common RNN architectures, LSTM and GRU, for a variety of tasks. In addition to the evaluation metric for each task, we report the number of RNN state updates (i.e., the number of elements in the input sequence used by the model) and the number of floating point operations (FLOPs) as measures of the computational load for each model. Since skipping an RNN update results in ignoring its corresponding input, we will refer to the number of updates and the number of used samples (i.e. elements in a sequence) interchangeably. With the goal of studying the effect of skipping state updates on the learning capability of the networks, we also introduce a baseline which skips a state update with probability $p_{skip}$. We tune the skipping probability to obtain models that perform a similar number of state updates to the Skip RNN models.

Training is performed with Adam (Kingma & Ba, 2014), learning rate of $10^{-4}$, $\beta_1 = 0.9$, $\beta_2 = 0.999$ and $\epsilon = 10^{-8}$ on batches of 256. Gradient clipping (Pascanu et al., 2013) with a threshold of 1 is applied to all trainable variables. Bias $b_p$ in Equation 4 is initialized to 1, so that all samples are used at the beginning of training[2]. The initial hidden state $s_0$ is learned during training, whereas $\tilde{u}_0$ is set to a constant value of 1 in order to force the first update at $t = 1$.

Experiments are implemented with TensorFlow[3] and run on a single NVIDIA K80 GPU.

### 4.1 ADDING TASK

We revisit one of the original LSTM tasks (Hochreiter & Schmidhuber, 1997), where the network is given a sequence of *(value, marker)* tuples. The desired output is the addition of only two values that are marked with a 1, whereas those marked with a 0 need to be ignored. We follow the experimental setup in Neil et al. (2016), where the first marker is randomly placed among the first 10% of samples (drawn with uniform probability) and the second one is placed among the last half of samples (drawn with uniform probability). This marker distribution yields sequences where at least 40% of the samples are distractors and provide no useful information at all. However, it is worth noting that in this task the risk of missing a marker is very large as compared to the benefits of working on shorter subsequences.

---

[2]In practice, forcing the network to use all samples at the beginning of training improves its robustness against random initializations of its weights and increases the reproducibility of the presented experiments. A similar behavior was observed in other augmented RNN architectures such as Neural Stacks (Grefenstette et al., 2015).

[3]https://www.tensorflow.org

| Model | Task solved | State updates | Inference FLOPs |
|---|---|---|---|
| LSTM | Yes | $100.0\% \pm 0.0\%$ | $2.46 \times 10^6$ |
| LSTM ($p_{skip} = 0.2$) | No | $80.0\% \pm 0.1\%$ | $1.97 \times 10^6$ |
| LSTM ($p_{skip} = 0.5$) | No | $50.1\% \pm 0.1\%$ | $1.23 \times 10^6$ |
| Skip LSTM, $\lambda = 0$ | Yes | $81.1\% \pm 3.6\%$ | $2.00 \times 10^6$ |
| Skip LSTM, $\lambda = 10^{-5}$ | Yes | $53.9\% \pm 2.1\%$ | $1.33 \times 10^6$ |
| GRU | Yes | $100.0\% \pm 0.0\%$ | $1.85 \times 10^6$ |
| GRU ($p_{skip} = 0.02$) | No | $98.0\% \pm 0.0\%$ | $1.81 \times 10^6$ |
| GRU ($p_{skip} = 0.5$) | No | $49.9\% \pm 0.6\%$ | $9.25 \times 10^5$ |
| Skip GRU, $\lambda = 0$ | Yes | $97.9\% \pm 3.2\%$ | $1.81 \times 10^6$ |
| Skip GRU, $\lambda = 10^{-5}$ | Yes | $50.7\% \pm 2.6\%$ | $9.40 \times 10^5$ |

Table 1: Results for the adding task, displayed as $mean \pm std$ over four different runs. The task is considered to be solved if the MSE is at least two orders of magnitude below the variance of the output distribution.

We train RNN models with 110 units each on sequences of length 50, where the values are uniformly drawn from $\mathcal{U}(-0.5, 0.5)$. The final RNN state is fed to a fully connected layer that regresses the scalar output. The model is trained to minimize the Mean Squared Error (MSE) between the output and the ground truth. We consider that a model is able to solve the task when its MSE on a held-out set of examples is at least two orders of magnitude below the variance of the output distribution. This criterion is a stricter version of the one followed by Hochreiter & Schmidhuber (1997).

While all models learn to solve the task, results in Table 1 show that Skip RNN models are able to do so with roughly half of the updates of their corresponding counterparts. We observed that the models using fewer updates never miss any marker, since the penalization in terms of MSE would be very large (see Section B.1 for examples). This is confirmed by the poor performance of the baselines that randomly skip state updates, which are not able to solve the tasks even when the skipping probability is low. Skip RNN models learn to skip most of the samples in the 40% of the sequence where there are no markers. Moreover, most updates are skipped once the second marker is found, since all the relevant information in the sequence has already been seen. This last pattern provides evidence that the proposed models effectively learn whether to update or copy the hidden state based on the input sequence, as opposed to learning biases in the dataset only. As a downside, Skip RNN models show some difficulties skipping a large number of updates at once, probably due to the cumulative nature of $\tilde{u}_t$.

## 4.2 MNIST CLASSIFICATION FROM A SEQUENCE OF PIXELS

The MNIST handwritten digits classification benchmark (LeCun et al., 1998) is traditionally addressed with Convolutional Neural Networks (CNNs) that efficiently exploit spatial dependencies through weight sharing. By flattening the $28 \times 28$ images into 784-d vectors, however, it can be reformulated as a challenging task for RNNs where long term dependencies need to be leveraged (Le et al., 2015b). We follow the standard data split and set aside 5,000 training samples for validation purposes. After processing all pixels with an RNN with 110 units, the last hidden state is fed into a linear classifier predicting the digit class. All models are trained for 600 epochs to minimize cross-entropy loss.

Table 2 summarizes classification results on the test set after 600 epochs of training. Skip RNNs are not only able to solve the task using fewer updates than their counterparts, but also show a lower variation among runs and train faster (see Figure 2). We hypothesize that skipping updates make the Skip RNNs work on shorter subsequences, simplifying the optimization process and allowing the networks to capture long term dependencies more easily. A similar behavior was observed for Phased LSTM, where increasing the sparsity of cell updates accelerates training for very long sequences (Neil et al., 2016). However, the drop in performance observed in the models where the state updates are skipped randomly suggests that learning which samples to use is a key component in the performance of Skip RNN.

| Model | Accuracy | State updates | Inference FLOPs |
|---|---|---|---|
| LSTM | $0.910 \pm 0.045$ | $784.00 \pm 0.00$ | $3.83 \times 10^7$ |
| LSTM ($p_{skip} = 0.5$) | $0.893 \pm 0.003$ | $392.03 \pm 0.05$ | $1.91 \times 10^7$ |
| Skip LSTM, $\lambda = 10^{-4}$ | $0.973 \pm 0.002$ | $379.38 \pm 33.09$ | $1.86 \times 10^7$ |
| GRU | $0.968 \pm 0.013$ | $784.00 \pm 0.00$ | $2.87 \times 10^7$ |
| GRU ($p_{skip} = 0.5$) | $0.912 \pm 0.004$ | $391.86 \pm 0.14$ | $1.44 \times 10^7$ |
| Skip GRU, $\lambda = 10^{-4}$ | $0.976 \pm 0.003$ | $392.62 \pm 26.48$ | $1.44 \times 10^7$ |
| TANH-RNN (Le et al., 2015a) | 0.350 | 784.00 | - |
| *i*RNN (Le et al., 2015a) | 0.970 | 784.00 | - |
| *u*RNN (Arjovsky et al., 2016) | 0.951 | 784.00 | - |
| *s*TANH-RNN (Zhang et al., 2016) | 0.981 | 784.00 | - |
| LSTM (Cooijmans et al., 2017) | 0.989 | 784.00 | - |
| BN-LSTM (Cooijmans et al., 2017) | 0.990 | 784.00 | - |

Table 2: Accuracy, used samples and average FLOPs per sequence at inference on the test set of MNIST after 600 epochs of training. Results are displayed as $mean \pm std$ over four different runs.

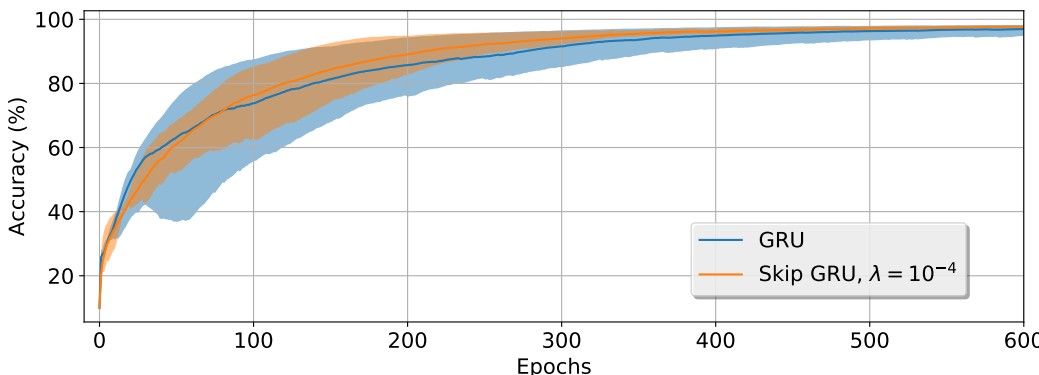

Figure 2: Accuracy evolution during training on the validation set of MNIST. The Skip GRU exhibits lower variance and faster convergence than the baseline GRU. A similar behavior is observed for LSTM and Skip LSTM, but omitted for clarity. Shading shows maximum and minimum over 4 runs, while dark lines indicate the mean.

The performance of RNN models on this task can be boosted through techniques like recurrent batch normalization (Cooijmans et al., 2017) or recurrent skip coefficients (Zhang et al., 2016). Cooijmans et al. (2017) show how an LSTM with specific weight initialization schemes for improved gradient flow (Le et al., 2015a; Arjovsky et al., 2016) can reach accuracy rates of up to $0.989\%$. Note that these techniques are orthogonal to skipping state updates and Skip RNN models could benefit from them as well.

Sequences of pixels can be reshaped back into 2D images, allowing to visualize the samples used by the RNNs as a sort of hard visual attention model (Xu et al., 2015). Examples such as the ones depicted in Figure 3 show how the model learns to skip pixels that are not discriminative, such as the padding regions in the top and bottom of images. Similarly to the qualitative results for the adding task (Section 4.1), attended samples vary depending on the particular input being given to the network.

### 4.3 TEMPORAL ACTION LOCALIZATION ON CHARADES

One popular approach to video analysis tasks today is to extract frame-level features with a CNN and modeling temporal dynamics with an RNN (Donahue et al., 2015; Yue-Hei Ng et al., 2015). Videos are commonly recorded at high sampling rates, generating long sequences with strong tem-

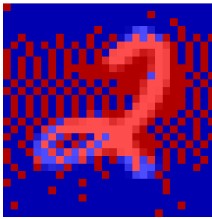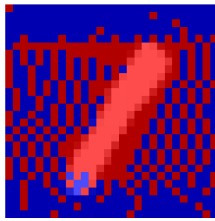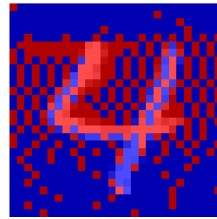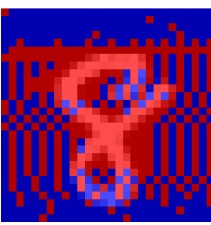

Figure 3: Sample usage examples for the Skip LSTM with $\lambda = 10^{-4}$ on the test set of MNIST. Red pixels are used, whereas blue ones are skipped.

poral redundancies that are challenging for RNNs. Moreover, processing frames with a CNN is computationally expensive and may become prohibitive for high frame rates. These issues have been alleviated in previous works by using short clips (Donahue et al., 2015) or by downsampling the original data in order to cover long temporal spans without increasing the sequence length excessively (Yue-Hei Ng et al., 2015). Instead of addressing the long sequence problem at the input data level, we let the network learn which frames need to be used.

Charades (Sigurdsson et al., 2016a) is a dataset containing 9,848 videos annotated for 157 action classes in a per-frame fashion. Frames are encoded using *fc7* features from the RGB stream of a Two-Stream CNN provided by the organizers of the challenge[4], extracted at 6 fps. The encoded frames are fed into two stacked RNN layers with 256 units each and the hidden state in the last RNN layer is used to compute the update probability for the Skip RNN models. Since each frame may be annotated with zero or more classes, the networks are trained to minimize element-wise binary cross-entropy at every time step. Unlike the previous sequence tagging tasks, this setup allows us to evaluate the performance of Skip RNN on a task where the output is a sequence as well.

Evaluation is performed following the setup by Sigurdsson et al. (2016c), but evaluating on 100 equally spaced frames instead of 25, and results are reported in Table 3. It is surprising that the GRU baselines that randomly skip state updates perform on par with their Skip GRU counterparts for low skipping probabilities. We hypothesize several reasons for this behavior, which was not observed in previous experiments: (1) there is a supervision signal at every time step and the inputs and (2) outputs are strongly correlated in consecutive frames. On the other hand, Skip RNN models clearly outperform the random methods when fewer updates are allowed. Note that this setup is far more challenging because of the longer time spans between updates, so properly distributing the state updates along the sequence is key to the performance of the models. Interestingly, Skip RNN models learn which frames need to be attended from RGB data and without having access to explicit motion information.

Skip GRU tends to perform fewer state updates than Skip LSTM when the cost per sample is low or none. This behavior is the opposite of the one observed in the adding task (Section 4.1), which may be related to the observation that determining the best performing gated unit depends on the task at hand Chung et al. (2014). Indeed, GRU models consistently outperform LSTM ones on this task. This mismatch in the number of used samples is not observed for large values of $\lambda$, as both Skip LSTM and Skip GRU converge to a comparable number of used samples.

A previous work reports better action localization performance by integrating RGB and optical flow information as an input to an LSTM, reaching $9.60\%$ mAP (Sigurdsson et al., 2016c). This boost in performance comes at the cost of roughly doubling the number of FLOPs and memory footprint of the CNN encoder, plus requiring the extraction of flow information during a preprocessing step. Interestingly, our model learns which frames need to be attended from RGB data and without having access to explicit motion information.

---

[4]http://vuchallenge.org/charades.html

| Model | mAP (%) | State updates | Inference FLOPs |
|---|---|---|---|
| LSTM | 8.40 | $172.9 \pm 47.4$ | $2.65 \times 10^{12}$ |
| LSTM ($p_{skip} = 0.75$) | 8.11 | $43.3 \pm 13.2$ | $6.63 \times 10^{11}$ |
| LSTM ($p_{skip} = 0.90$) | 7.21 | $17.2 \pm 6.1$ | $2.65 \times 10^{11}$ |
| Skip LSTM, $\lambda = 0$ | 8.32 | $172.9 \pm 47.4$ | $2.65 \times 10^{12}$ |
| Skip LSTM, $\lambda = 10^{-4}$ | 8.61 | $172.9 \pm 47.4$ | $2.65 \times 10^{12}$ |
| Skip LSTM, $\lambda = 10^{-3}$ | 8.32 | $41.9 \pm 11.3$ | $6.41 \times 10^{11}$ |
| Skip LSTM, $\lambda = 10^{-2}$ | 7.86 | $17.4 \pm 4.4$ | $2.66 \times 10^{11}$ |
| GRU | 8.70 | $172.9 \pm 47.4$ | $2.65 \times 10^{12}$ |
| GRU ($p_{skip} = 0.10$) | 8.94 | $155.6 \pm 42.9$ | $2.39 \times 10^{12}$ |
| GRU ($p_{skip} = 0.40$) | 8.81 | $103.6 \pm 29.3$ | $1.06 \times 10^{12}$ |
| GRU ($p_{skip} = 0.70$) | 8.42 | $51.9 \pm 15.4$ | $7.95 \times 10^{11}$ |
| GRU ($p_{skip} = 0.90$) | 7.09 | $17.3 \pm 6.3$ | $2.65 \times 10^{11}$ |
| Skip GRU, $\lambda = 0$ | 8.94 | $159.9 \pm 46.9$ | $2.45 \times 10^{12}$ |
| Skip GRU, $\lambda = 10^{-4}$ | 8.76 | $100.8 \pm 28.1$ | $1.54 \times 10^{12}$ |
| Skip GRU, $\lambda = 10^{-3}$ | 8.68 | $54.2 \pm 16.2$ | $8.29 \times 10^{11}$ |
| Skip GRU, $\lambda = 10^{-2}$ | 7.95 | $18.4 \pm 5.1$ | $2.82 \times 10^{11}$ |

Table 3: Mean Average Precision (mAP), used samples and average FLOPs per sequence at inference on the validation set of Charades. The number of state updates is displayed as $mean \pm std$ over all the videos in the validation set.

## 5 CONCLUSION

We presented Skip RNNs as an extension to existing recurrent architectures enabling them to skip state updates thereby reducing the number of sequential operations in the computation graph. Unlike other approaches, all parameters in Skip RNN are trained with backpropagation. Experiments conducted with LSTMs and GRUs showed that Skip RNNs can match or in some cases even outperform the baseline models while relaxing their computational requirements. Skip RNNs provide faster and more stable training for long sequences and complex models, owing to gradients being backpropagated through fewer time steps resulting in a simpler optimization task. Moreover, the introduced computational savings are better suited for modern hardware than those methods that reduce the amount of computation required at each time step (Koutnik et al., 2014; Neil et al., 2016; Chung et al., 2017).

### ACKNOWLEDGMENTS

This work was partially supported by the Spanish Ministry of Economy and Competitivity and the European Regional Development Fund (ERDF) under contracts TEC2016-75976-R and TIN2015-65316-P, by the BSC-CNS Severo Ochoa program SEV-2015-0493, and grant 2014-SGR-1051 by the Catalan Government. Víctor Campos was supported by Obra Social "la Caixa" through La Caixa-Severo Ochoa International Doctoral Fellowship program. We would also like to thank the technical support team at the Barcelona Supercomputing Center.

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

## A    ADDITIONAL EXPERIMENTS

### A.1    FREQUENCY DISCRIMINATION TASK

In this experiment, the network is trained to classify between sinusoids whose period is in range $T \sim \mathcal{U}(5, 6)$ milliseconds and those whose period is in range $T \sim \{(1, 5) \cup (6, 100)\}$ milliseconds (Neil et al., 2016). Every sine wave with period $T$ has a random phase shift drawn from $\mathcal{U}(0, T)$. At every time step, the input to the network is a single scalar representing the amplitude of the signal. Since sinusoid are continuous signals, this tasks allows to study whether Skip RNNs converge to the same solutions when their parameters are fixed but the sampling period is changed. We study two different sampling periods, $T_s = \{0.5, 1\}$ milliseconds, for each set of hyperparameters.

We train RNNs with 110 units each on input signals of 100 milliseconds. Batches are stratified, containing the same number of samples for each class, yielding a 50% chance accuracy. The last state of the RNN is fed into a 2-way classifier and trained with cross-entropy loss. We consider that a model is able to solve the task when it achieves an accuracy over 99% on a held-out set of examples.

Table 4 summarizes results for this task. When no cost per sample is set ($\lambda = 0$), the number of updates differ under different sampling conditions. We attribute this behavior to the potentially large number of local minima in the cost function, since there are numerous subsampling patterns for which the task can be successfully solved and we are not explicitly encouraging the network to converge to a particular solution. On the other hand, when $\lambda > 0$ Skip RNN models with the same cost per sample use roughly the same number of input samples even when the sampling frequency is doubled. This is a desirable property, since solutions are robust to oversampled input signals. Qualitative results can be found in Section B.2.

| Model | $\mathbf{T_s = 1ms}$ (length 100) | | $\mathbf{T_s = 0.5ms}$ (length 200) | |
|---|---|---|---|---|
| | Task solved | State updates | Task solved | State updates |
| LSTM | Yes | $100.0 \pm 0.00$ | Yes | $200.0 \pm 0.00$ |
| Skip LSTM, $\lambda = 0$ | Yes | $55.5 \pm 16.9$ | Yes | $147.9 \pm 27.0$ |
| Skip LSTM, $\lambda = 10^{-5}$ | Yes | $47.4 \pm 14.1$ | Yes | $50.7 \pm 16.8$ |
| Skip LSTM, $\lambda = 10^{-4}$ | Yes | $12.7 \pm 0.5$ | Yes | $19.9 \pm 1.5$ |
| GRU | Yes | $100.0 \pm 0.00$ | Yes | $200.0 \pm 0.00$ |
| Skip GRU, $\lambda = 0$ | Yes | $73.7 \pm 17.9$ | Yes | $167.0 \pm 18.3$ |
| Skip GRU, $\lambda = 10^{-5}$ | Yes | $51.9 \pm 10.2$ | Yes | $54.2 \pm 4.4$ |
| Skip GRU, $\lambda = 10^{-4}$ | Yes | $23.5 \pm 6.2$ | Yes | $22.5 \pm 2.1$ |

Table 4: Results for the frequency discrimination task, displayed as $mean \pm std$ over four different runs. The task is considered to be solved if the classification accuracy is over 99%. Models with the same cost per sample ($\lambda > 0$) converge to a similar number of used samples under different sampling conditions.

### A.2    SENTIMENT ANALYSIS ON IMDB

The IMDB dataset (Maas et al., 2011) contains 25,000 training and 25,000 testing movie reviews annotated into two classes, *positive* and *negative* sentiment, with an approximate average length of 240 words per review. We set aside 15% of training data for validation purposes. Words are embedded into 300-d vector representations before being fed to an RNN with 128 units. The embedding matrix is initialized using pre-trained word2vec[5] embeddings (Mikolov et al., 2013) when available, or random vectors drawn from $\mathcal{U}(-0.25, 0.25)$ otherwise (Kim, 2014). Dropout with rate $0.2$ is applied between the last RNN state and the classification layer in order to reduce overfitting. We evaluate the models on sequences of length 200 and 400 by cropping longer sequences and padding shorter ones (Yu et al., 2017).

Results on the test are reported in Table 5. In a task where it is hard to predict which input tokens will be discriminative, the Skip RNN models are able to achieve similar accuracy rates to the baseline

---

[5]https://code.google.com/archive/p/word2vec/

models while reducing the number of required updates. These results highlight the trade-off between accuracy and the available computational budget, since a larger cost per sample results in lower accuracies. However, allowing the network to select which samples to use instead of cropping sequences at a given length boosts performance, as observed for the Skip LSTM (length 400, $\lambda = 10^{-4}$), which achieves a higher accuracy than the baseline LSTM (length 200) while seeing roughly the same number of words per review. A similar behavior can be seen for the Skip RNN models with $\lambda = 10^{-3}$, where allowing them to select words from longer reviews boosts classification accuracy while using a comparable number of tokens per sequence.

In order to reduce overfitting of large models, Miyato et al. (2017) leverage additional unlabeled data through adversarial training and achieve a state of the art accuracy of $0.941$ on IMDB. For an extended analysis on how different experimental setups affect the performance of RNNs on this task, we refer the reader to (Longpre et al., 2016).

| Model | Length 200 | | Length 400 | |
|---|---|---|---|---|
| | Accuracy | State updates | Accuracy | State updates |
| LSTM | $0.843 \pm 0.003$ | $200.00 \pm 0.00$ | $0.868 \pm 0.004$ | $400.00 \pm 0.00$ |
| Skip LSTM, $\lambda = 0$ | $0.844 \pm 0.004$ | $196.75 \pm 5.63$ | $0.866 \pm 0.004$ | $369.70 \pm 19.35$ |
| Skip LSTM, $\lambda = 10^{-5}$ | $0.846 \pm 0.004$ | $197.15 \pm 3.16$ | $0.865 \pm 0.001$ | $380.62 \pm 18.20$ |
| Skip LSTM, $\lambda = 10^{-4}$ | $0.837 \pm 0.006$ | $164.65 \pm 8.67$ | $0.862 \pm 0.003$ | $186.30 \pm 25.72$ |
| Skip LSTM, $\lambda = 10^{-3}$ | $0.811 \pm 0.007$ | $73.85 \pm 1.90$ | $0.836 \pm 0.007$ | $84.22 \pm 1.98$ |
| GRU | $0.845 \pm 0.006$ | $200.00 \pm 0.00$ | $0.862 \pm 0.003$ | $400.00 \pm 0.00$ |
| Skip GRU, $\lambda = 0$ | $0.848 \pm 0.002$ | $200.00 \pm 0.00$ | $0.866 \pm 0.002$ | $399.02 \pm 1.69$ |
| Skip GRU, $\lambda = 10^{-5}$ | $0.842 \pm 0.005$ | $199.25 \pm 1.30$ | $0.862 \pm 0.008$ | $398.00 \pm 2.06$ |
| Skip GRU, $\lambda = 10^{-4}$ | $0.834 \pm 0.006$ | $180.97 \pm 8.90$ | $0.853 \pm 0.011$ | $314.30 \pm 2.82$ |
| Skip GRU, $\lambda = 10^{-3}$ | $0.800 \pm 0.007$ | $106.15 \pm 37.92$ | $0.814 \pm 0.005$ | $99.12 \pm 2.69$ |

Table 5: Accuracy and used samples on the test set of IMDB for different sequence lengths. Results are displayed as $mean \pm std$ over four different runs.

### A.3 ACTION CLASSIFICATION ON UCF-101

UCF-101 (Soomro et al., 2012) is a dataset containing 13,320 trimmed videos belonging to 101 different action categories. We use 10 seconds of video sampled at 25 fps, cropping longer ones and padding shorter examples with empty frames. Activations in the Global Average Pooling layer from a ResNet-50 (He et al., 2016) CNN pretrained on the ImageNet dataset (Deng et al., 2009) are used as frame-level features, which are fed into two stacked RNN layers with 512 units each. The weights in the CNN are not tuned during training to reduce overfitting. The hidden state in the last RNN layer is used to compute the update probability for the Skip RNN models.

We evaluate the different models on the first split of UCF-101 and report results in Table 6. Skip RNN models do not only improve the classification accuracy with respect to the baseline, but require very few updates to do so, possibly due to the low motion between consecutive frames resulting in frame-level features with high temporal redundancy (Shelhamer et al., 2016). Moreover, Figure 4 shows how models performing fewer updates converge faster thanks to the gradients being preserved during longer spans when training with backpropagation through time.

Non recurrent architectures for video action recognition that have achieved high performance on UCF-101 comprise CNNs with spatiotemporal kernels (Tran et al., 2015) or two-stream CNNs (Simonyan & Zisserman, 2014). Carreira & Zisserman (2017) show the benefits of expanding 2D CNN filters into 3D and pretraining on larger datasets, obtaining an accuracy of $0.845$ when using RGB data only and $0.934$ when incorporating optical flow information.

| Model | Accuracy | State updates | Inference FLOPs |
|---|---|---|---|
| LSTM | 0.671 | 250.0 | $9.52 \times 10^{11}$ |
| Skip LSTM, $\lambda = 0$ | 0.749 | 138.9 | $5.29 \times 10^{11}$ |
| Skip LSTM, $\lambda = 10^{-5}$ | 0.757 | 24.2 | $9.21 \times 10^{10}$ |
| Skip LSTM, $\lambda = 10^{-4}$ | 0.790 | 7.6 | $2.89 \times 10^{10}$ |
| GRU | 0.791 | 250.0 | $9.51 \times 10^{11}$ |
| Skip GRU, $\lambda = 0$ | 0.796 | 124.2 | $4.73 \times 10^{11}$ |
| Skip GRU, $\lambda = 10^{-5}$ | 0.792 | 29.7 | $1.13 \times 10^{11}$ |
| Skip GRU, $\lambda = 10^{-4}$ | 0.793 | 23.7 | $9.02 \times 10^{10}$ |
| I3D (RGB) (Carreira & Zisserman, 2017) | 0.845 | - | - |
| Two-stream I3D (Carreira & Zisserman, 2017) | 0.934 | - | - |

Table 6: Accuracy, used samples and average FLOPs per sequence at inference on the validation set of UCF-101 (split 1).

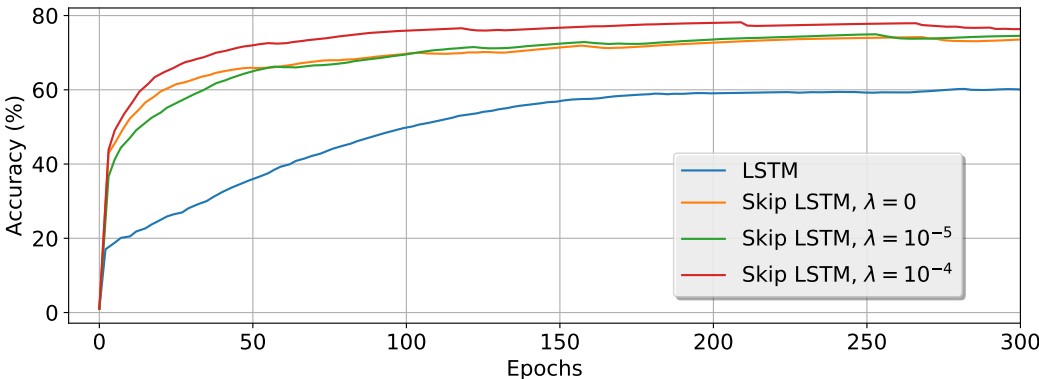

Figure 4: Accuracy evolution during the first 300 training epochs on the validation set of UCF-101 (split 1). Skip LSTM models converge much faster than the baseline LSTM.

# B    QUALITATIVE RESULTS

This appendix contains additional qualitative results for the Skip RNN models.

## B.1    ADDING TASK

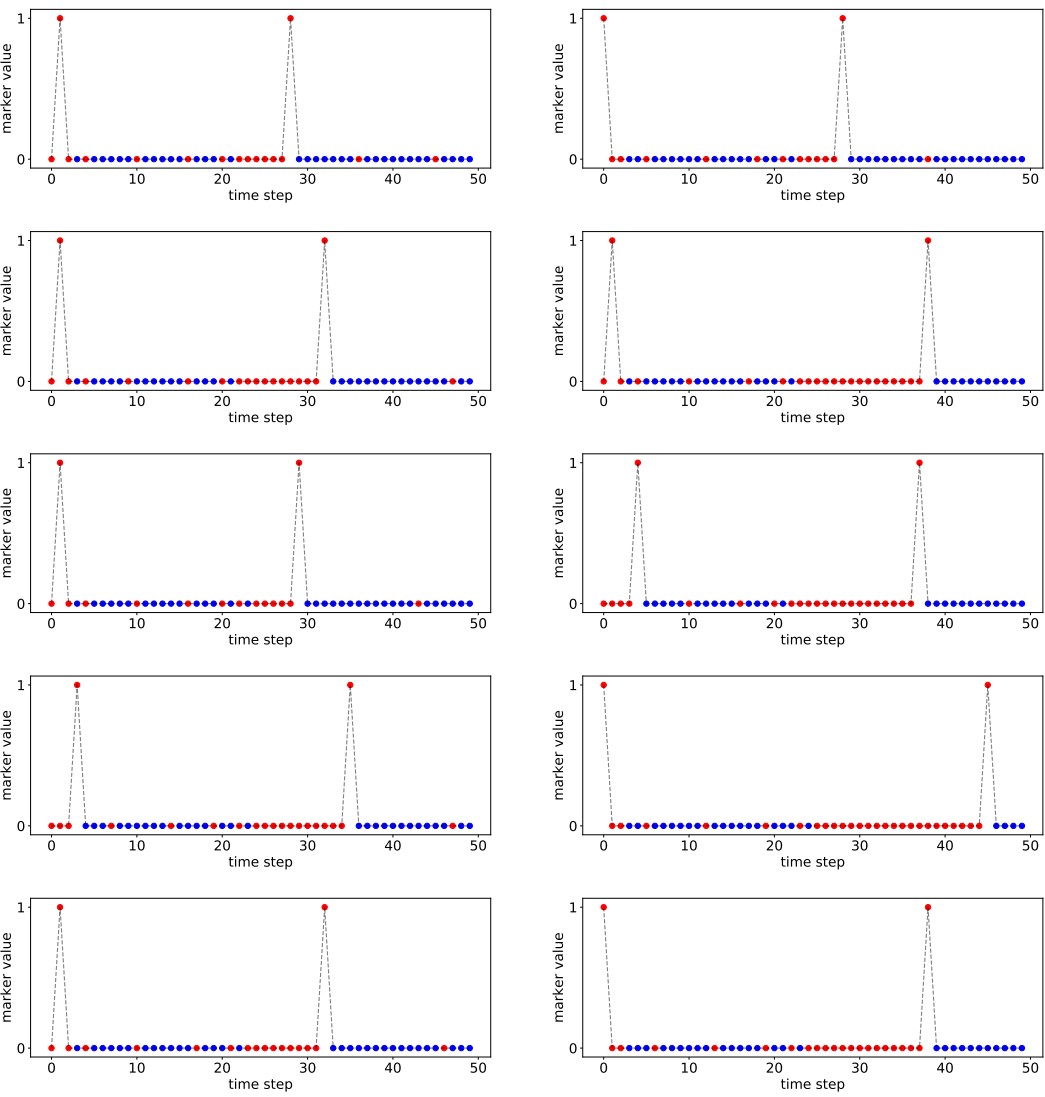

Figure 5: Sample usage examples for the Skip GRU with $\lambda = 10^{-5}$ on the adding task. Red dots indicate used samples, whereas blue ones are skipped.

## B.2 FREQUENCY DISCRIMINATION TASK

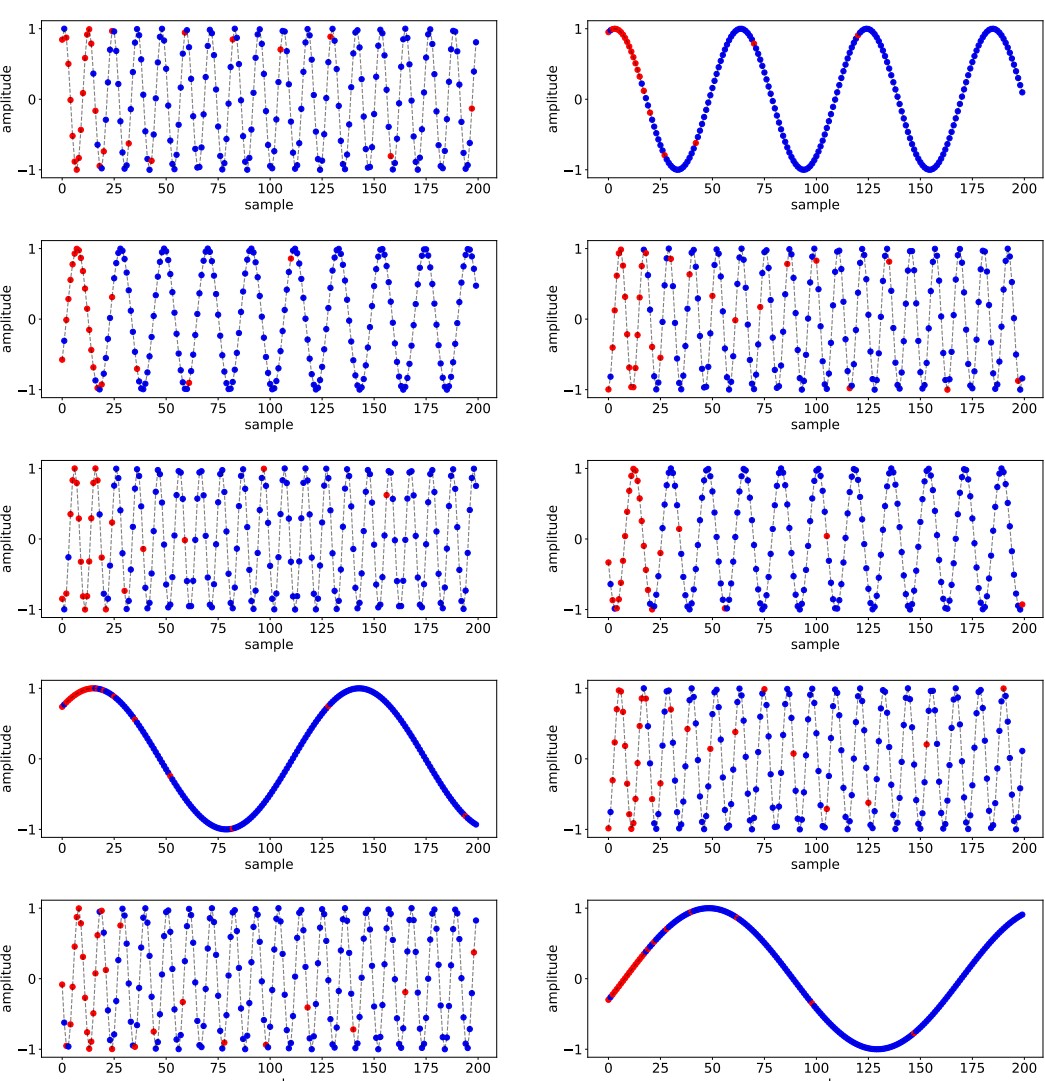

Figure 6: Sample usage examples for the Skip LSTM with $\lambda = 10^{-4}$ on the frequency discrimination task with $T_s = 0.5$ms. Red dots indicate used samples, whereas blue ones are skipped. The network learns that using the first samples is enough to classify the frequency of the sine waves, in contrast to a uniform downsampling that may result in aliasing.

