# OpenReview forum: "Skip RNN: Learning to Skip State Updates in Recurrent Neural Networks"
_ICLR.cc/2018/Conference — Accept (Poster)_

### Official Review · AnonReviewer1 · 2017-11-26
**Interesting model, but lacking comparison to prior work**

**Rating:** 6
**Confidence:** 4

**Review:**

UPDATE: Following the author's response I've increased my score from 5 to 6. The revised paper includes many of the additional references that I suggested, and the author response clarified my confusion over the Charades experiments; their results are indeed close to state-of-the-art on Charades activity localization (slightly outperformed by [6]), which I had mistakenly confused with activity classification (from [5]).

The paper proposes the Skip RNN model which allows a recurrent network to selectively skip updating its hidden state for some inputs, leading to reduced computation at test-time. At each timestep the model emits an update probability; if this probability is over a threshold then the next input and state update will be skipped. The use of a straight-through estimator allows the model to be trained with standard backpropagation. The number of state updates that the model learns to use can be controlled with an auxiliary loss function. Experiments are performed on a variety of tasks, demonstrating that the Skip-RNN compares as well or better than baselines even when skipping nearly half its state updates.

Pros:
- Task of reducing computation by skipping inputs is interesting
- Model is novel and interesting
- Experiments on multiple tasks and datasets confirm the efficacy of the method
- Skipping behavior can be controlled via an auxiliary loss term
- Paper is clearly written

Cons:
- Missing comparison to prior work on sequential MNIST
- Low performance on Charades dataset, no comparison to prior work
- No comparison to prior work on IMDB Sentiment Analysis or UCF-101 activity classification

The task of reducing computation by skipping RNN inputs is interesting, and the proposed method is novel, interesting, and clearly explained. Experimental results across a variety of tasks are convincing; in all tasks the Skip-RNNs achieve their goal of performing as well or better than equivalent non-skipping variants. The use of an auxiliary loss to control the number of state updates is interesting; since it sometimes improves performance it appears to have some regularizing effect on the model in addition to controlling the trade-off between speed and accuracy.

However, where possible experiments should compare directly with prior published results on these tasks; none of the experiments from the main paper or supplementary material report any numbers from any other published work.

On permuted MNIST, Table 2 could include results from [1-4]. Of particular interest is [3], which reports 98.9% accuracy with a 100-unit LSTM initialized with orthogonal and identity weight matrices; this is significantly higher than all reported results for the sequential MNIST task.

For Charades, all reported results appear significantly lower than the baseline methods reported in [5] and [6] with no explanation. All methods work on “fc7 features from the RGB stream of a two-stream CNN provided by the organizers of the [Charades] challenge”, and the best-performing method (Skip GRU) achieves 9.02 mAP. This is significantly lower than the two-stream results from [5] (11.9 mAP and 14.3 mAP) and also lower than pretrained AlexNet features averaged over 30 frames and classified with a linear SVM, which [5] reports as achieving 11.3 mAP. I don’t expect to see state-of-the-art performance on Charades; the point of the experiment is to demonstrate that Skip-RNNs perform as well or better than their non-skipping counterparts, which it does. However I am surprised at the low absolute performance of all reported results, and would appreciate if the authors could help to clarify whether this is due to differences in experimental setup or something else.

In a similar vein, from the supplementary material, sentiment analysis on IMDB and action classification on UCF-101 are well-studied problems, but the authors do not compare with any previously published results on these tasks.

Though experiments may not show show state-of-the-art performance, I think that they still serve to demonstrate the utility of the Skip-RNN architecture when compared side-by-side with a similarly tuned non-skipping baseline. However I feel that the authors should include some discussion of other published results.

On the whole I believe that the task and method are interesting, and experiments convincingly demonstrate the utility of Skip-RNNs compared to the author’s own baselines. I will happily upgrade my rating of the paper if the authors can address my concerns over prior work in the experiments.


References

[1] Le et al, “A Simple Way to Initialize Recurrent Networks of Rectified Linear Units”, arXiv 2015
[2] Arjovsky et al, “Unitary Evolution Recurrent Neural Networks”, ICML 2016
[3] Cooijmans et al, “Recurrent Batch Normalization”, ICLR 2017
[4] Zhang et al, “Architectural Complexity Measures of Recurrent Neural Networks”, NIPS 2016
[5] Sigurdsson et al, “Hollywood in homes: Crowdsourcing data collection for activity understanding”, ECCV 2016
[6] Sigurdsson et al, “Asynchronous temporal fields for action recognition”, CVPR 2017

---

> ### Author Response · Authors · 2017-12-20
> **Reply to AnonReviewer1**
>
> Q: Include prior published results of the same tasks
>
> A: Following the suggestion from the reviewer, we added previously published results and their implementation context  for all tasks mentioned by the reviewer: MNIST, Charades, UCF-101 and IMDB. They were added to the tables when possible, or to the discussion only otherwise (e.g. results from other works wouldn’t match the table layout for IMDB).
>
>
> Q: low performance on Charades dataset
>
> A: Regarding the performance in temporal action localization task on Charades, we would like to highlight that we report results for the action localization task (i.e. a many-to-many task where an output is emitted for each input frame). However, the results reported in [5] belong to the action classification task (i.e. a many-to-one task where a single output is emitted for the whole video). To the best of our knowledge, the best result for a CNN+LSTM architecture like ours on the localization task is 9.60% (Table 2 in [6], results without post-processing). However, in [6] they use both streams from a Two-Stream CNN (RGB+Flow) whereas we use the RGB stream only. Although this yields a 0.58% mAP increase with respect to our best performing model, using both streams results in approximately a 2x increase in computation (# FLOPs) and memory footprint, plus an additional pre-processing step to compute the optical flow from the input frames. We are not aware of any other work using RGB only with CNN+LSTM on this dataset and task. It is also interesting to notice that our models learn which frames need to be attended without being given explicit motion information as input.

---

> > ### Comment · AnonReviewer1 · 2018-01-22
> > **repy**
> >
> > I appreciate the author's response and updates to the paper, and I apologize for my confusion over action classification vs action localization; this explanation makes the Charades experiments more convincing, and I've increased my score from 5 to 6 accordingly.

---

### Official Review · AnonReviewer3 · 2017-11-27
**interesting idea. requires more experiment to be convincing.**

**Rating:** 6
**Confidence:** 4

**Review:**

This paper proposes an idea to do faster RNN inference via skip RNN state updates.
I like the idea of the paper, in particular the design which enables calculating the number of steps to skip in advance. But the experiments are not convincing enough. First the tasks it was tested on are very simple -- 2 synthetic tasks plus 1 small-scaled task. I'd like to see the idea works on larger scale problems -- as that is where the computation/speed matters. Also besides the number of updates reported in table, I think the wall-clock time for inference should also be reported, to demonstrate what the paper is trying to claim.

Minor --
Cite Estimating or Propagating Gradients Through Stochastic Neurons for Conditional Computation by Yoshua Bengio, Nicholas Leonard and Aaron Courville for straight-through estimator.

---

> ### Author Response · Authors · 2017-12-20
> **Reply to AnonReviewer3**
>
> Q: Wall-clock time for inferencing should be reported.
>
> A: Wall-clock timing is highly dependent on factors such as hardware, framework and implementation, thus making it difficult to isolate the impact of the model. This is why we originally reported the number of sequential steps (i.e. state updates) performed by each model. As an alternative to wall-clock timing, we updated the manuscript to report the number of floating point operations (FLOPs) per sequence. This measure is also independent of external factors such as hardware/software while being representative of the computational load of each model. Although we believe that wall-clock time is not very informative, we are willing to report it if the reviewer still thinks that it will improve the quality of the paper.
>
>
>
> Q: Cite paper by Bengio et al.
>
> A: The paper by Bengio et al. is cited for the ST estimator in the updated version of the manuscript.

---

### Official Review · AnonReviewer2 · 2017-11-28

**Rating:** 6
**Confidence:** 4

**Review:**

The authors proposed a novel RNN model where both the input and the state update of the recurrent cells are skipped adaptively for some time steps. The proposed models are learned by imposing a soft constraint on the computational budget to encourage skipping redundant input time steps. The experiments in the paper demonstrated skip RNNs outperformed regular LSTMs and GRUs o thee addition, pixel MNIST and video action recognition tasks.



Strength:
- The experimental results on the simple skip RNNs have shown a good improvement over the previous results.

Weakness:
- Although the paper shows that skip RNN worked well, I found the appropriate baseline is lacking here. Comparable baselines, I believe, are regular LSTM/GRU whose inputs are randomly dropped out during training.

- Most of the experiments in the main paper are on toy tasks with small LSTMs. I thought the main selling point of the method is the computational gain. Would it make more sense to show that on large RNNs with thousands of hidden units? After going over the additional experiments in the appendix, and I find the three results shown in the main paper seem cherry-picked, and it will be good to include more NLP tasks.

---

> ### Author Response · Authors · 2017-12-20
> **Reply to AnonReviewer 2**
>
> Q: Comparison with baselines randomly dropping inputs is missing.
>
> A: We actually have reported the random input dropping baseline for the MNIST task. In the revised version of the paper, we have added results of random dropping baseline for the adding task and Charades. In all cases, the proposed model learns the best ways to skip states (instead randomly) and demonstrated clear performance gains over the random dropping baseline. Note here we assume when random dropping is done, both the input and the state update operation are skipped. We do not consider the option that only input is dropped and the state is still updated since it does not achieve the objective of saving computation.
>
>
>
> Q: the three experiments included in the main paper seemed cherry-picked.
>
> A: We have included a large diverse set of experiments in the Appendix including signal frequency discrimination, sentiment analysis on IMDB movie reviews (text), and video action classification. Far from cherry-picking test results, our goal is to demonstrate the general applicability of the proposed model in various tasks involving data of different modalities (signal, text, and video). We will be happy to move any of the experiments from Appendix to the main paper.

---

> > ### Comment · AnonReviewer2 · 2018-01-22
> > **reply**
> >
> > I would like to thank the authors for their reply. The new experiments with randomly dropout baseline look compiling. My only concern is the performance of the random baseline in Table 3 is as good as the best skip GRU regarding mAP. The latest revision has cleared some of my concerns in the initial review. I decided to increase the review score from 5 to 6.

---

### Author Response · Authors · 2017-12-20
**On the scale of the experiments & updates to the paper**

We thank reviewers for their valuable comments. We respond to the main concerns below and in individual replies to each reviewer.

R2 and R3
Q: Task scale and model size too small. Should run experiments with a large number of hidden units.

A:
- SkipRNN has indeed a larger advantage over regular RNNs when the model is larger, since the computational savings of skipped states are larger. However, this does not necessarily require RNNs with thousands of hidden units, as the main bulk of computation may come from other associated components of the full architecture, e.g. the CNN encoder in video tasks. It’s important to note when a RNN state update is skipped for a time step, all its preceding elements in the computation graph for that time step are also skipped. For example, in the case of CNN encoder in video tasks, the computational cost of these elements is typically very significant.  The updated version of the manuscript reports the actual # of FLOPs/sequence and shows how SkipRNN models can result in large savings in computation-intensive tasks such as video action localization. We estimated the # of floating point operations based on the actual operations involved in the input encoder and the RNN model.

- Please also note that the size of the studied RNNs in our paper is the same as or even larger than those reported in related methods, e.g. [1, 2, 3]. The largest model in these works is composed by 2 LSTM layers with 256 units each, while we have reported results for 2 layers with 512 units each in the appendix.
We believe that the size of the considered tasks is also comparable to those in [1, 2, 3]. Despite some of them using larger datasets in terms of number of examples, their inputs have low dimensionality (e.g. 300-d pre-trained word embeddings) compared to the ones in some of our experiments (e.g. up to 4096-d for video tasks).



Updates to the paper:

- Add FLOPs to the tables
- Moved the description of the random skipping baseline to the beginning of the experiments section
- Add skipping baselines for the adding task (plus discussion)
- Add skipping baselines for Charades (plus discussion)
- Evaluate models on Charades sampling 100 frames/video instead of 25, which should be more accurate for studying models performing different number of state updates.
- Add SOTA results for recurrent models on MNIST
- Add comparison to Sigurdsson et al. (CVPR 2017) for Charades
- Cite prior work and SOTA for IMDB
- Add SOTA results on UCF-101 (Carreira & Zisserman, 2017)




[1] Neil et al., Phased LSTM: Accelerating Recurrent Network Training for Long or Event-based Sequences, NIPS 2017
[2] Yu et al., Learning to Skim Text, ACL 2017
[3] Anonymous authors, Neural Speed Reading via Skim-RNN, ICLR 2018 submission

---

### Decision · Program_Chairs · 2018-01-29
**ICLR 2018 Conference Acceptance Decision**

**Decision:**

Accept (Poster)

**Comment:**

This paper explores what might be characterized as an adaptive form of ZoneOut.
With the improvements and clarifications added to the paper during the rebuttal the paper could be accepted.